



# Automatic Quality Control of the Meteosat First Generation Measurements

Freek Liefhebber[1], Sarah Lammens[1], Paul Brussee[1], André Bos[1], Viju O. John[2], Frank Rüthrich[2], Jacobus Onderwaater[2], Michael G. Grant[2], and Jörg Schulz[2]

[1]Science and Technology B.V., Delft, The Netherlands
[2]EUMETSAT, Darmstadt, Germany

**Correspondence:** Freek Liefhebber (liefhebber@stcorp.nl)

**Abstract.** Now that the Earth has been monitored by satellites for more than 40 years, Earth Observation images can be used to study how the Earth system behaves over extended periods. Such long-term studies require the combination of data from multiple instruments, with the earliest data sets being of particular importance in establishing a baseline for trend analysis. As the quality of these earlier datasets is often lower, careful quality control is essential, but the sheer size of these image sets makes an inspection by hand impracticable. Therefore, one needs to resort to automatic methods to inspect these Earth Observation images for anomalies. In this paper, we describe the design of a system that performs an automatic anomaly analysis on Earth Observation images, in particular the Meteosat first generation measurements. The design of this system is based on a preliminary analysis of the typical anomalies that can be found in the data set. This preliminary analysis was conducted by hand on a representative subset and resulted in a finite list of anomalies that needed to be detected in the whole data set. The automated anomaly detection system employs a dedicated detection algorithm for each of these anomalies. The result is a system with a high probability of detection and low false alarm rate. Furthermore, most of these algorithms are able to pinpoint the anomalies to the specific pixels affected in the image, allowing the maximum use of the data available.

## 1 Introduction

Earth Observation (EO) from geostationary satellites provides a wealth of information which can be used to study the Earth's climate system as described, for example, by Rossow and Schiffer (1999). While their focus of interest were long term cloud effects, other studies also have used those data for deriving information on land surface temperatures (Duguay-Tetzlaff et al. (2015)), upper tropospheric humidity (Soden and Bretherton (1993)) and solar irradiation (Mueller et al. (2015)). EUMETSAT's Meteosat First Generation (MFG) satellites were equipped with the Meteosat Visible Infra-Red Imager (MVIRI) instrument. The MVIRIs had been measuring in three distinct wavelengths: i) visible (VIS) - providing information on surface and atmospheric albedo (John and Soden (2007); Ruethrich et al. (2019)); ii) water vapour (WV) providing information on upper tropospheric humidity, which is a key climate variable, yet not very well simulated by the current climate models Stoeckli et al. (2019); and iii) infrared (IR) window providing information on the surface and cloud top temperature and on the presence of clouds John et al. (2019). Data from Meteosat-1, launched in 1977, are available for only a year from December 1978 to



November 1979. There are continuous data available from Meteosat-2 onwards starting in February 1982 until April 2017 when the last satellite in the series, Meteosat-7, was moved to its graveyard orbit.

If such an image dataset is to be used for analysing how the climate varies over time, one must control the quality of the dataset in order to avoid any bias in the result. The potential value of such a historical time series, however, is partially hindered due to the presence of image anomalies (Koepken (2004)) (i.e. acquisition errors, physical effects such as straylight, etc) and radiometric anomalies (Brogniez et al. (2006)) (i.e. calibration errors, calibration drift, etc). There have been considerable efforts in recent years to correct for radiometric anomalies in these measurements (Ruethrich et al. (2019); John et al. (2019)), but so far there were no attempts to detect and correct for image anomalies. For the MFG satellites, EUMETSAT keeps an archive of the level 1.0 data (raw images with geolocation tie points but prior to rectification and calibration) and level 1.5 data (rectified to a fixed geolocation grid and calibrated). These data were archived in near real-time, meaning no corrections have been applied beyond that in the original processing. For a planned reprocessing of the level 1.0 to level 1.5 data addressing data anomalies, it is mandatory to create a consistent and as-complete-as-possible set of information concerning each Meteosat image including its metadata and the detection and flagging of anomalies in the measurements. An image anomaly is defined as an anomaly in the radiometric content of an image not caused by the rectification process, a satellite manoeuvre, scheduled change in satellite parameters (decontamination, gain changes). If such unexpected image radiometric anomalies occur, they can have a very detrimental impact on the use of the images. Except for problems such as wrong gain settings, wrong channel configuration, they will usually occur within the commissioning phase of a new satellite or in the period after just taking up operations with a new satellite. However, they still can occur suddenly on an operational satellite due to system failure. Another type of anomaly can be found in the metadata of the images. Due to various reasons, including operator error and control software problems, the metadata can be inconsistent or incomplete with respect to the scientific contents of the images. This paper tackles the issue of image anomalies in the MVIRI measurements by processing the whole archive of MVIRI images to detect and flag any anomalies present.

The paper is organized as follows: Section 2 briefly describes the MVIRI data and known anomalies in them, Section 3 presents the algorithm to detect and flag such anomalies, Section 4 discusses the results, and Section 5 provides conclusions and outlook.

## 2   Meteosat First Generation Measurements

During the lifetime of Meteosat First Generation series, the satellites were operated at different orbit locations. The primary 0° longitude orbit position has been covered from the start, supplemented by the so-called Indian Ocean Data Coverage service (IODC) from 1998 onwards with MFG satellites located at 57°E and 63°E. IODC has been continued with the first Meteosat Second Generation (MSG) satellite operating from 41.5°E. Furthermore, in the first half of the 1990s, the Meteosat-3 satellite was moved to the West Atlantic to support coverage of the United States of America, thereby providing the so-called Atlantic Data Coverage (ADC) and the Extended Atlantic Data Coverage (XADC) services from orbit positions at 50°W and 75°W, respectively. Some of these satellites also were operated in rapid scanning mode (RSS) - the details are shown in Table 1.



**Table 1.** List of satellite names, operational mission with nominal sub-satellite longitude position in brackets, and the main years of operation.

| Satellite | Mission | Main Operational Years | Number of level 1.0 files |
|---|---|---|---|
| Meteosat-2 | 0-degree (0°) | 1981-1988 | 110195 |
| Meteosat-3 | 0-degree (0°) | 1988-1991 | 89614 |
| Meteosat-3 | ADC (50°W) | 1991-1993 | |
| Meteosat-3 | XADC (75°W) | 1993-1995 | |
| Meteosat-4 | 0-degree (0°) | 1989-1994 | 75728 |
| Meteosat-5 | 0-degree (0°) | 1991-1997 | 210260 |
| Meteosat-5 | 0-degree RSS (0°) | 1997-1998 | |
| Meteosat-5 | IODC (63°E) | 1998-2007 | |
| Meteosat-6 | 0-degree (0°) | 1996-1998 | 144773 |
| Meteosat-6 | 0-degree RSS (0°) | 2000-2007 | |
| Meteosat-6 | IODC (67°E) | 2007-2009 | |
| Meteosat-7 | 0-degree (0°) | 1998-2006 | 329334 |
| Meteosat-7 | IODC (57°E) | 2006-2017 | |

The METEOSAT Visible Infra-Red Imager (MVIRI) instrument measures in three spectral channels (Table 2): Visible, Water Vapour (WV), and Thermal Infrared (TIR). The VIS channel has 2 detectors, VIS-South and VIS-North. There are 48 acquisition slots in a day (one every 30 minutes) and within one slot a MVIRI scan is created. A MVIRI scan consists of 3030 scanlines, where 2500 scanlines belong to the image-acquiring forward scan. The VIS-South and VIS-North detectors create 5 5000 samples along each scanline and the TIR- and WV-detectors create 2500 samples along each scanline.

**Table 2.** Spatial and spectral characteristics of MVIRI visible (VIS), thermal infrared (TIR), water vapour (WV) channels.

| Channel | res. nadir (km) | Nominal spectral band ($\mu$m) |
|---|---|---|
| VIS 0.7 | 2.5 | 0.40 - 1.10 |
| WV 6.4 | 5.0 | 5.70 - 7.10 |
| TIR 11.5 | 5.0 | 10.5 - 12.5 |

## 3 Anomaly Detection Methods

Anomaly detection can be approached in various ways, ranging from machine-learning techniques to image processing in combination with classification techniques (Hodge and Austin (2004)). Each approach has its own merits, such as the effort to implement the anomaly detection system, the performance of the detection process, and the trust the end-users ultimately have 10 in the approach selected. The implementation effort depends on the different types of anomalies one can expect in the dataset, and whether it is possible to limit the number of anomaly types. If there is not a good understanding of what types of anomaly



one can expect, it will be difficult to program the algorithms that detect the presence of an anomaly in an image, but one can better concentrate on algorithms that define the nominal case in order to detect deviations from the norm. However, such an algorithm will be limited in its capability to classify or identifying the type of anomalies. The performance of an anomaly detector can be described in terms of the Probability of Detection (PoD), the False Alarm Rate (FAR), and the specificity

of the detection (i.e. whether the anomaly can be isolated to a subset of affected pixels rather than discarding the whole image). Finally, the trust that the ultimate end-users have depends primarily on how much one can understand the workings of the algorithm and whether there is some indication by the algorithm as to why an image is classified as anomalous (e.g. by providing an overview of the affected pixels, and the type of anomaly).

A machine-learning approach is appealing from an implementation effort point of view, but in this case, the more classical

approach of image processing using a manually selected array of anomaly classification algorithms has been used. The benefit of this well-known approach is that, during the investigation and development process, knowledge is generated on the exact appearance of the various anomaly types, which lead to improved end-user trust and a better chance of identifying the source of errors. There is also more control to tune the quality of the anomaly detection of the analysed image, especially in terms of specificity, and to prevent overtraining of an algorithm on a limited dataset. The remainder of this section describes the process

of analysing a limited test set of data to identify anomalies and tune algorithms, and improving the quality of the analysis. We call this subset of representative images the *training set*. It should be understood that this training set is used for visual inspection and to learn about the possible anomalies, and not for the automatic training of machine-learning algorithms. Next, several anomaly categories are presented and the development process of the detection algorithm will be discussed. Finally, to provide a feel for the type of algorithms applied, three anomaly detection concepts will be briefly described.

## 3.1 Creating the training-set

During the development of the MFG satellites and the years of operation, valuable knowledge on all kinds of topics related to the satellite and its application has been created. This knowledge can be related to typical low-level sensors aspects, such as signal-to-noise ratio or crosstalk, but also to the data-processing of the sensed data and the applied corrections. For the MFG satellites, the active development period was several decades ago and the involved scientists and engineers are largely not

available anymore. As a result, the present knowledge on anomalies in the dataset due to malfunctioning sensors or incorrect application is limited and diminishing with time. However, this lack of background information is also an opportunity to analyse the historical dataset from first principles instead of only focusing on known anomalies.

Therefore, the first step was a manual study of anomalies in the historical dataset. Due to the high number of images (around 1 million), only a limited subset can be manually examined. The objective was to find a representative set of anomaly types that

one could expect in the whole data set. Therefore, a training dataset was created with representative coverage of the relevant channels, periods and satellites. Specifically, samples were randomly selected but with several constraints in order to avoid large gaps in time or that a satellite, channel or typical timeslot (e.g. January 1st on 12:00h) is over represented. The dataset should be as small as possible for practicality reasons, but also large enough to cover all anomaly types. As most anomaly types (e.g. straylight effect) will affect multiple channels in the same timeslot, the dataset contains only one channel from any





particular time slot, such that the size of dataset is kept small while maximising the number of independent images searched for anomalies. The main interest is in anomalies related to sensor failure or radiometric effects present in level 1.0 data, but it is also possible that the level 1.5 rectification process could introduce new anomalies, and thus the training-set dataset consists of both level 1.0 and level 1.5 images. This dataset was inspected manually to characterise:

- Types of anomalies

- Frequency of occurrence

- Appearance and severity of an anomaly

- Origin of the anomaly / root cause

Each image from the dataset is evaluated to determine if it contains an anomaly. The dataset together with the evaluation is
called the "training-set" and is later used to tune and evaluate the performance of the automatic detection software. The manual inspection process requires several iterations to converge on consistent human decision criteria for flagging images. A difficult aspect is that the appearance of an anomaly in an image or for a particular pixel is modelled as an effect that is present or is not present (i.e. it cannot be partially present). In several cases, the severity of the appearance of the anomaly is small and it is doubtful whether an image or a pixel should be flagged.

**3.2   Improving the quality of the training-set**

The manual inspection of the MVIRI images is a difficult and a time consuming job. A dedicated tool has been developed to speed up and to improve the quality of this manual inspection process. The inspection process also determines which anomaly types exist and how they look like, which is a learning process. To avoid inconsistent (human) judgements during this learning process, the images are inspected in several iterations. The manual inspection has been executed with greatest care, but also
we also have to conclude that the quality of a human-inspected dataset is lower than desired. Common mistakes include inconsistent detection accuracy, mistakes concerning anomaly types and missed pixels. On the other hand, humans are very good at detecting patterns and abnormalities in images, which would probably not be detected by any algorithm with very limited a priori information. To achieve the highest quality possible (no missed cases or false cases), an iterative strategy was chosen where the manual inspection and evaluation is corrected by algorithms (see Figure 1). With this strategy, the training-
set is initially defined by manual inspection. New anomaly cases are detected by algorithms based on the initial evaluation and, after manual inspection and confirmation, the new cases are added to the training-set. The algorithms used to detect new cases can be all kinds of detection algorithms, but include the to-be-developed anomaly detection algorithms. In our case, the algorithms used to correct the manual inspection were a combination of the final automatic detection algorithms and other (generic) detection algorithms. An example of generic detection algorithm is one that detects if the average intensity and the
standard deviation are within a certain range. The usage of a detection algorithm is especially useful when the appearance of a particular anomaly cannot be visually detected by a human in a single image. The accuracy of such generic detection algorithms can be quite low, but they help with flagging images where the severity of the anomaly is quite low. An example in





the literature of such a generic detection algorithm in use was to detect cases of the "loose cold optics" issue of the Meteosat 6 satellite (Holmlund (2005)). The proposed method has low detection accuracy but, with manual inspection and evaluation of the algorithm's internal calculations, the quality of the training-set can be increased. The strategy of improving the quality of the dataset is applied during the entire development of the anomaly detection algorithms and during the evaluation of the detection performance.

## 3.3 Anomaly types

The training-set dataset was analysed and 30 different anomaly types were defined. The specifics of the defined anomaly types and their appearance will be unique to the MFG satellites, but the general anomaly origin or category will also hold for other similar geostationary imaging satellite types, such as GOES (Schmetz and Menzel (2015); Considine (2006)) and GMS/MTSAT (Tabata et al. (2019)). For the MFG anomalies, some examples of categories follow (see Table 3 for a complete overview):

- Missing or corrupt data (see Figure 2). All pixels of an image or a scanline have the value '0' or an obviously incorrect value.

- Low quality sensory data (see Figure 3. For example, the signal-to-noise ratio of the image is much lower than expected or the pixels are affected by a disturbance source.

- Unexpected behaviour of (historical) processing. The processing software used has changed during the lifetime of the satellites and these changes sometimes resulted in different behaviour. An example of unexpected behaviour is a different definition of the start time of a scan or that pixels have been set to '0' due to various different reasons.

- Stray light related anomalies (see Figure 4). Indirect illumination of a light detector by internal reflections; e.g. in the right locations, the sun will reflect off internal components of the telescope and onto pixels that are not looking at the sun. This stray light effect will directly affect the various detectors (VIS, IR and WV), but it can also indirectly affect the consecutive scans for reasons which are currently not known to us. With the MFG satellites, the WV-images were affected several hours after the initial stray light effect. Images have also been affected by stray light from the moon.

- Instable optics related anomalies. The Meteosat 5 and 6 satellites suffer from known hardware issues related to the optics, which has the effect that the sensitivity changes in time.

The training-set also contains level 1.5 images, but we have not discovered any anomaly that is related to the level 1.5 rectification process. In general, it holds that discovered anomalies in the level 1.5 images are better recognizable than in the level 1.0 images. The rectification process (Wolff (1985) blurs anomalies on the pixel level so that if an anomaly only affects a single scanline in the level 1.0 images, its effect in the level 1.5 image will be a blurry curved line.

Table 3 shows an overview of the defined anomaly types.





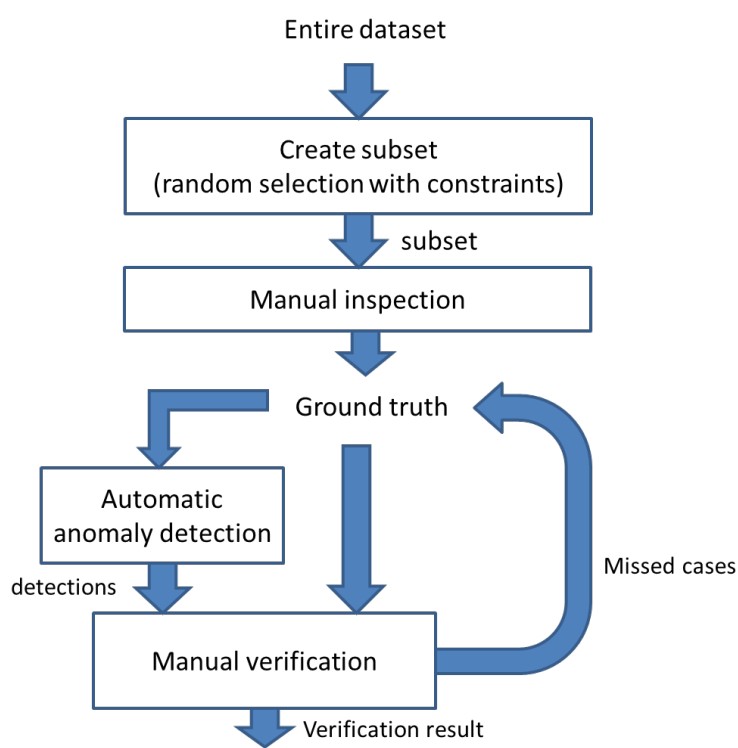

**Figure 1.** Approach for creating training-set, where a manually inspected image is corrected by algorithms.



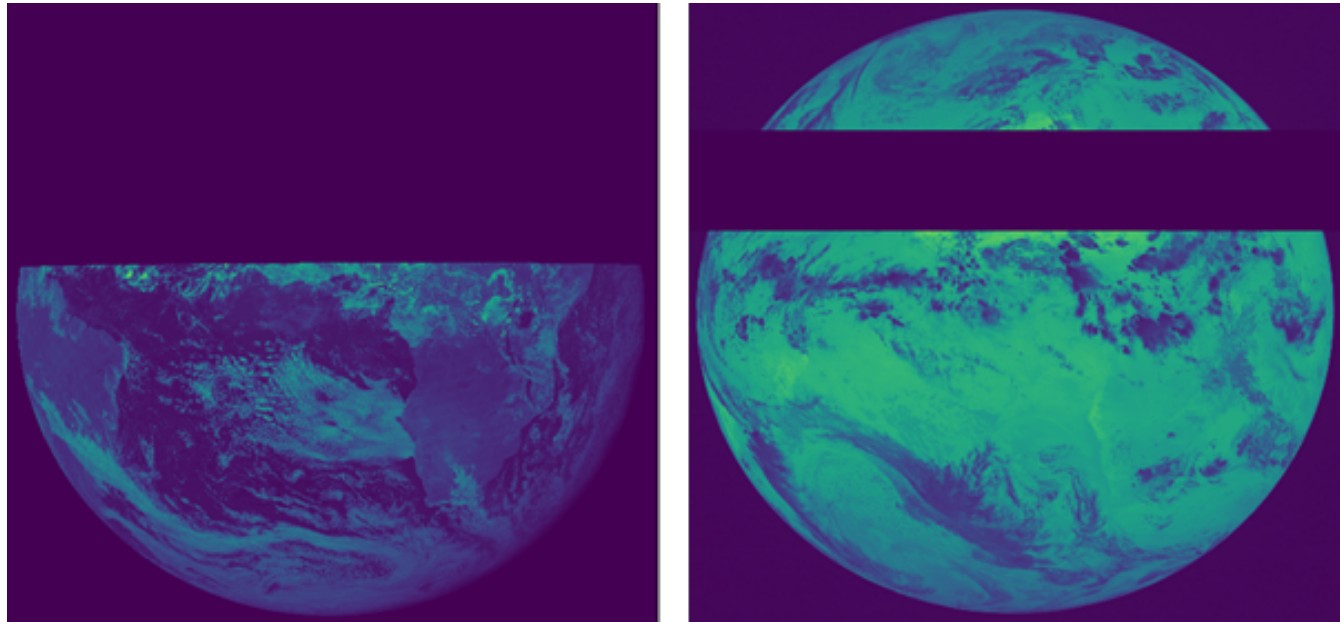

**Figure 2.** Examples of missing data.

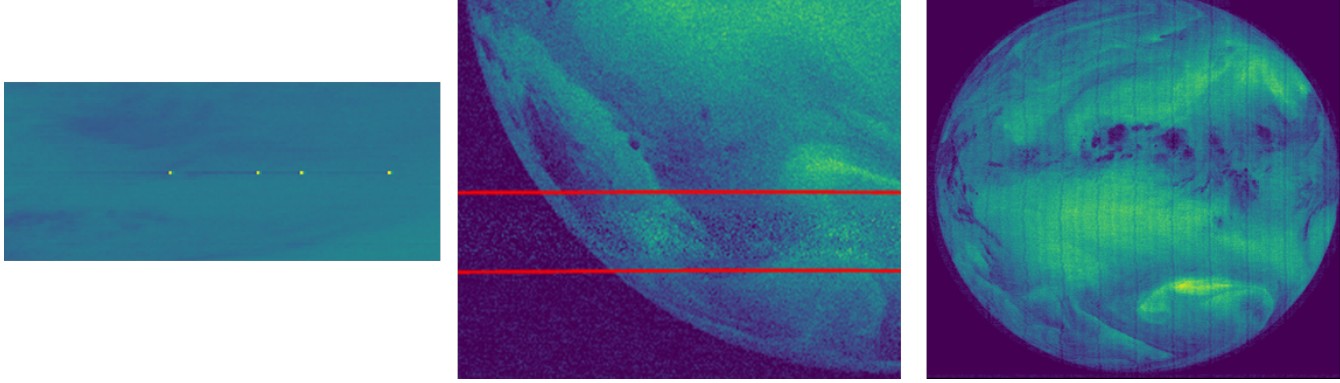

**Figure 3.** Examples of low quality sensory data. Left: a scanline contains an interference pattern. Center: a block of scanlines (inside the red lines) with a much lower signal-to-noise ratio than neighbouring scanlines. Right: image contains an unknown disturbance pattern.

**Table 3.** Description of the defined anomaly types in the MFG measurement

| CATEGORY | TYPE | Description |
|---|---|---|
| artefact | misalignment | Scanlines are not aligned properly and the east- and the west horizon of the Earth is not a continuous curve. |
| | over-illumination (overflow) | Over-illuminated pixels have an incorrect value of 124 instead of the maximum value of 255. |
| | tilted line | In a WV-image, a line under an angle 19 degrees (from the vertical) is visible. |
| celestial body | celestial body: the Moon | The Moon is present in the MVIRI-image. |
| | celestial body: undefined | An artefact appearing similar to a celestial body was detected in the space area of the image, but the moon can be excluded due to orbital position |
| corrupt or missing | completely black | All or almost all pixels of a channel have an intensity lower than 10 (a threshold significantly higher than background noise). |
| | corrupt file | The size of the level 0 file is too small to contain data of all scanlines. |
| | hanging scanline | Position of detector has not changed |
| | incomplete image | Forward scan time/length is too small to capture the entire Earth. Note that if a rapid scan image is analysed, the captured image is considered as incomplete because the software is currently only designed for full Earth images. |
| | invalid signal | According to the meta-data, a channel is invalid. |
| | large black area | The image contains a large black area, where several scanlines are completely black (intensity is 0). |
| | large white area | The image contains a large white area, where several scanlines are completely white (intensity is 255). |
| | no sub images | The scan does not contain any forward scan. |
| hot pixel | hot pixel pattern 1 | A typical pixel pattern on a single scanline, which only appears in one channel. |
| | hot pixel pattern 2 | A typical pixel pattern on a single scanline, which appears in all channels at the same position. |
| | hot pixel pattern independent | Randomly distributed hot pixels (high intensity and very unlikely compared to the neighbouring pixels). |
| instable optics | instable optics | The observed sensitivity of the detector is not constant in time. This anomaly mainly appears with the known optical hardware issues of Meteosat 5 and 6. |
| low SNR | low SNR: scanline | The observed noise level in a scanline is much higher than the observed noise level in the entire image. |
| meta data | EFF position corrupt | The stored position of the satellite in the Earth Fixed Frame format is corrupt. |
| | orbit position empty | The stored position of the satellite is empty. |
| | parameter empty | A meta data parameter, which should have a value, is empty or zero. |
| | start time: forward scan | The start time definition used is unexpected. Here, the start time of the "scan" is equal to start time of the forward scan. |
| | start time: southern horizon | The start time definition used is unexpected. Here, the start time of the "scan" is equal to the moment when the southern horizon is detected. |
| | start time: start image | The start time definition used is unexpected. Here, the start time of the "scan" is equal to the start time of the entire scan. |
| | start time: undefined | The definition of the start time cannot be determined. |
| | value unexpected | A meta data parameter has an unexpected value (outside a certain range). |
| Raw data manipulated | background noise removed | Raw data has been changed and pixels that should contain background noise have '0' as their value. |
| | background noise removed and noise added | Raw data has been changed. Besides the removal of the background noise, the intensity of all pixels might have been adjusted. |
| | the number of scanlines changed | The number of valid scanlines has been changed. |
| stray light | direct stray light | Image is affected by parasitic light via indirect optical path, which results in a typical pattern where the pixels have a higher intensity. |
| | indirect stray light | After the appearance of the direct stray light effect, several WV-images can be affected. With the affected WV-images, several scanlines will have a significant lower intensity. The affected scanlines with the indirect stray light anomaly are the scanlines, where in a preceding image the direct stray light effect was present. |
| suspicious pattern | reflection of the Moon | An over-illuminated, fingernail / crescent shaped blob. It often appears on the right side of the Moon. |
| | suspicious pattern | The WV-image contains a non-physical pattern. |


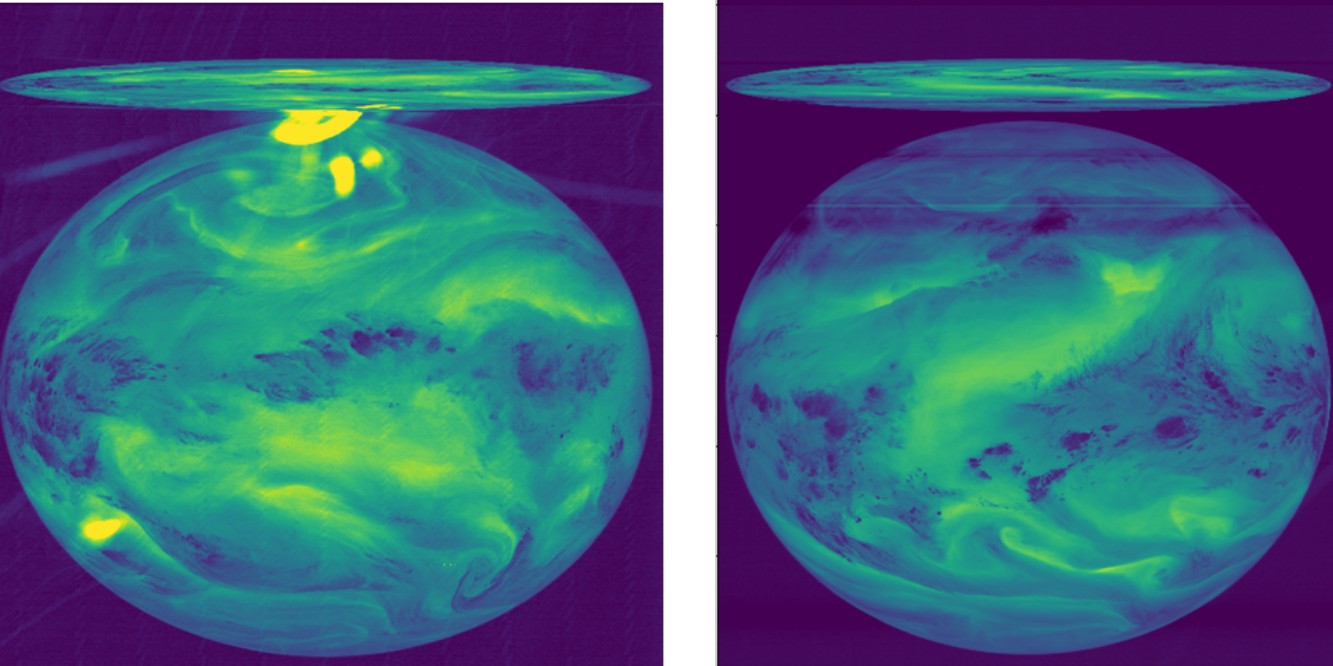

**Figure 4.** Left: image affected by a stray light anomaly. Right: several hours later, still showing follow-on effects after the stray light anomaly.

## 3.4 Development strategy anomaly detection algorithms

The creation of the training-set gave insight into the appearance and the probability of occurrence of the defined anomaly types. Some anomaly types occur very often, while others only had a few examples in the training-set. Some anomaly types are limited to a particular satellite and others are related to a particular channel or period of the day. The training-set covers more than

5 2500 timeslots, but still is far too small to calculate reliable probabilities of occurrence for several anomaly types. Therefore, during the development of the detection algorithms, this uncertainty in the probability of occurrence must be continually taken into account. For each anomaly type, the most likely root cause or origin of the anomaly is determined to avoid mistakes in the estimation of the probability of occurrence in a certain situation (channel, satellite, period of the day etc).

Distinctive features for specific anomalies can be various metadata parameters, such as satellite-id or time, but in general the
10 focus is on image-based parameters. The following data sources can be used for the detection of an anomaly:

– Metadata parameters of the file (satellite-id, date, time, geo-location etc)

– Image data of a channel

– Other channels of same timeslot

– Series of consecutive images





In general, it is preferable to minimise the required number of parameters examined for an anomaly because each parameter can be affected by other anomalies than the targeted one. Therefore, when multiple parameters are used, care must be taken to ensure that each parameter is genuinely necessary. Also, feature calculations are complicated by the fact that files (or parameters) may be missing; this mainly occurred when the majority of a scan is affected by the straylight effect. In general,

anomalies are more recognisable in the raw data (level 1.0 files), so a choice was made to do anomaly detection only on level 1.0 files. If the detection requires a series of consecutive images, the algorithm has to perform some kind of registration between the images. The level 1.0 file contains information for the alignment or registration, which is used for an approximated rectification based on a linear homography transform (transform is defined by a 3x3 matrix multiplication). The (internal) approximated rectification results are verified on the basis of a cross-correlation. If a "better" shift (X,Y-displacement) between images can

be found with cross-correlation, this improved shift is applied to register the images.

In addition to detecting the presence of an anomaly, the area it affects in the image needs to be determined. The affected area can be described on the following levels:

- Image-level, where the entire or majority of the image is affected by an anomaly.

- Scanline-level, where a single or multiple scanlines are affected by an anomaly.

- Pixel-level, where the anomaly affects a pixel or multiple pixels.

In general, the affected area was already calculated by the calculation of the distinctive features for anomaly detection. The affected area is stored in a database and, to keep the database efficient, the affected area (scanline or pixels) is approximated by a list of rectangles (each specified by X & Y coordinates of a corner plus X & Y size).

For each anomaly type, a dedicated algorithm needs to be developed. The algorithm development for some anomaly types,

such as the missing data anomaly (several scanlines are missing), is quite straightforward. Others, such as the more image-processing based anomalies, can be very challenging and require elaborated, innovative detection concepts. The next sections will briefly describe the detection concept of three difficult anomaly types, to illustrate the variety of processing steps used. It will mainly focus on the basic detection concept and not on all important pre-processing steps and details to reach robustness under all circumstances.

## 3.5 Example complex-anomaly detection: "suspicious pattern"

Figure 5 shows two examples that suffer from the "suspicious pattern" anomaly. This anomaly type may have various appearances (due to unknown root causes), which all result in a suspicious repeating pattern. The repetitive pattern is clearly visible, but the magnitude is still quite small. Such repetitive patterns can be detected by the analysis of the 2D FFT spectrum, where this anomaly will introduce peaks in the 2D FFT spectrum. As the variation in appearance is very large (magnitude, spatial sep-

aration of the repeats, vertical or horizontal pattern), we do not search for a particular pattern / peak but compare the observed 2D FFT spectrum with the expected 2D FFT spectrum.

To detect these peaks in the 2D FFT spectrum, we divide the observed 2D FFT from a single image by the expected 2D FFT from the particular satellite. The expected 2D FFT from a satellite is calculated by averaging the 2D FFT from 100 images that





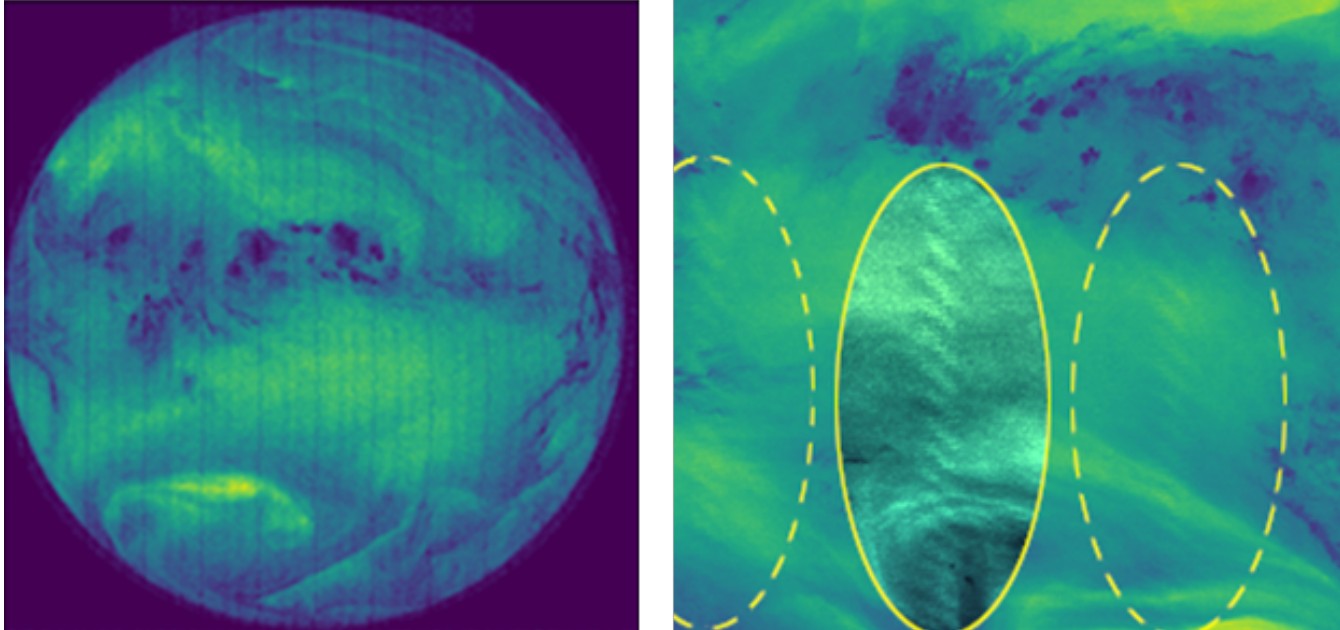

**Figure 5.** Two examples of the "suspicious pattern" anomaly: In the left image the vertical stripes are visible. In the right image the ovals indicate the position of the suspicious pattern; the middle oval has been altered to enhance visibility of the pattern.

did not contain any anomaly. If the ratio between the two 2D FFT spectra is larger than a threshold (count value of 10), we define it as a peak. The threshold has been manually determined with the aim to detect peaks in images where they would be identified also by human eye.

A (clear) peak in the 2D FFT spectrum does not always result in a noticeable (by humans) pattern in the spatial domain (normal image). This especially holds for peaks that correspond to fast changing patterns with a small magnitude (smaller than 1/255 of the maximum intensity). The effect of the detected FFT-peaks can be calculated by comparing the difference between the original and a reconstructed image. The reconstructed image is calculated by the inverse FFT of the 2D FFT spectrum with the peaks removed / reduced to a normal value. Only if this difference exceeds a threshold (T) will an anomaly be flagged. The flowchart of Figure 6 describes the process of detecting suspicious patterns in a schematic fashion.

## 3.6 Example complex-anomaly detection: "direct straylight"

The design of the MFG-satellites suffers from the issue that the detector can be illuminated via an indirect optical path, which results in the occurrence of the so-called direct straylight anomaly. This "parasitic" light causes a pattern with an increased intensity in the image. The observed pattern often contains characteristic bows or arcs, but its appearance is a little bit different in every instance. Figure 7 shows three example images affected by the direct straylight anomaly.





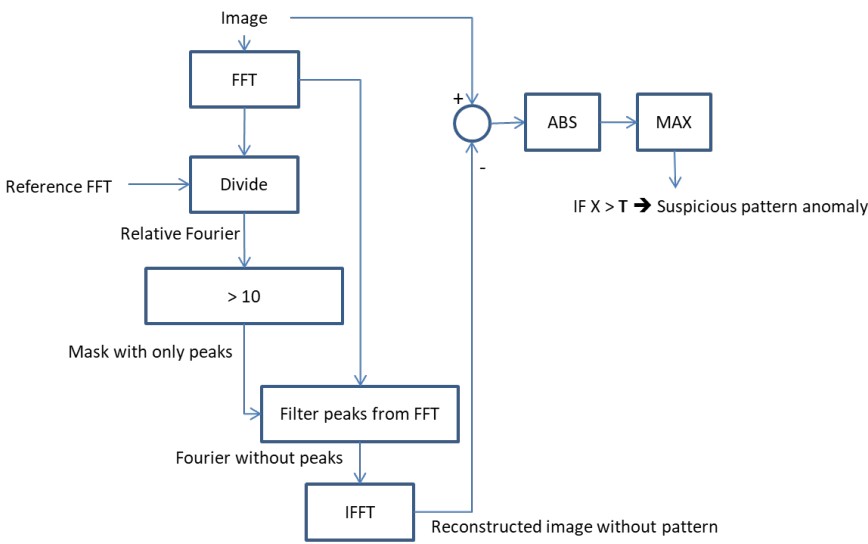

**Figure 6.** Conceptual processing pipeline for detecting the suspicious pattern anomaly.

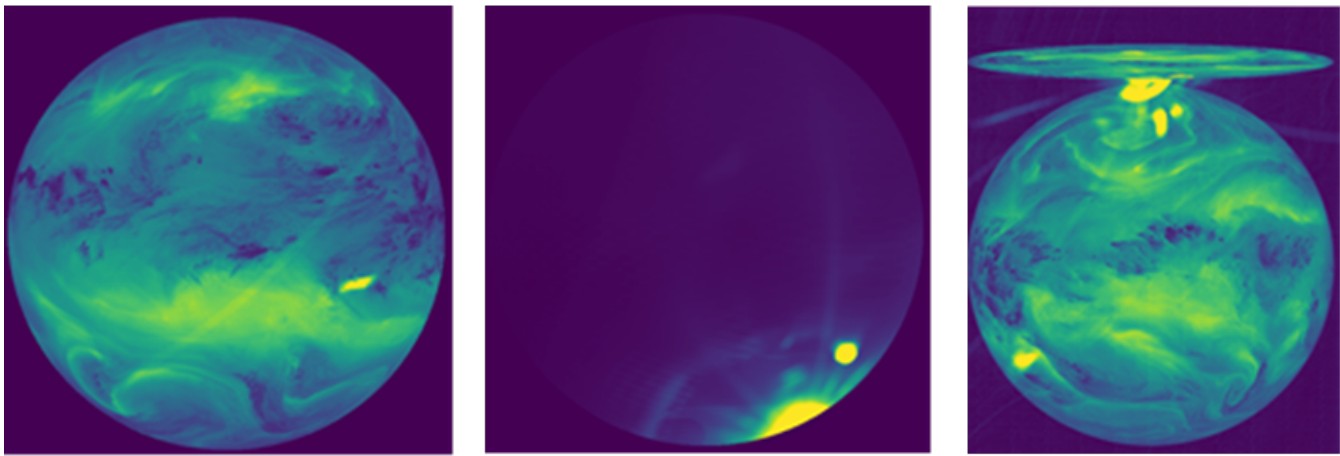

**Figure 7.** Three examples of the direct straylight effect anomaly.





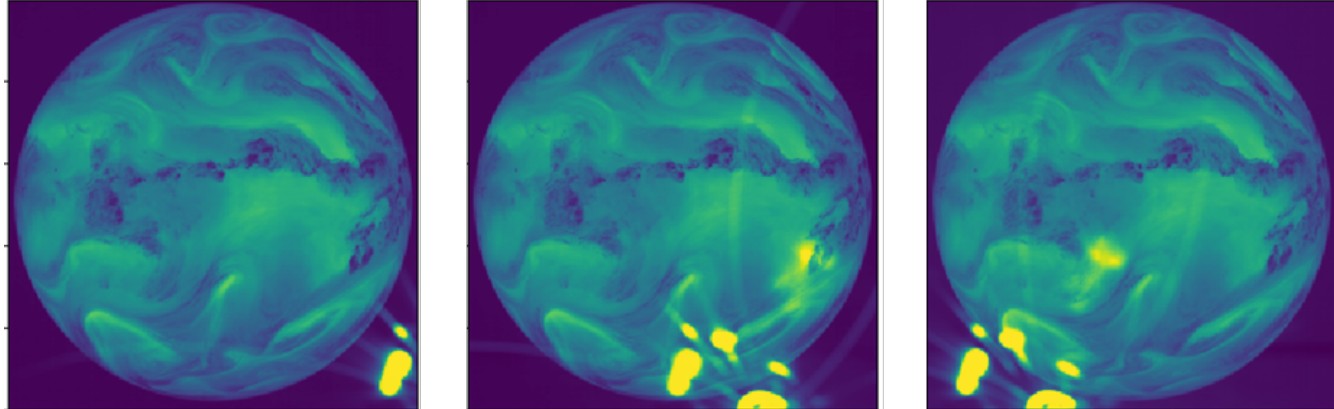

**Figure 8.** Three sequential images affected by the direct straylight anomaly.

In a series of images, the straylight pattern moves across the scene quite fast compared to the normal movements of the background (clouds, etc), which enables its detection. Figure 8 shows three sequential images that are affected by the same direct straylight anomaly. For detection, we assume that an affected pixel is (significantly) brighter than the same pixel of preceding and consecutive image.

To be able to use this assumption, it is essential that the raw images are aligned with each other and that no other anomalies have affected them. This allows the algorithm to determine if a pixel is affected by this anomaly (see Figure 9). Note that the bow in the lower left corner of the image has not been identified. The reason for this is that the algorithm compares the current image with the previous one. In this case the previous image also contained the same bow, and therefore the algorithm fails to identify the bow as anomalous.

**3.7    Example complex-anomaly detection: "instable optics"**

Meteosat 5 and 6 both suffer from a known optical hardware issue, which affects the sensitivity of the detector (Koepken (2004)). The sensitivity of the detector can vary by just a few percent and the effect is not noticeable by a human, which makes it hard to manually select images that are affected by the hardware issue. The detector's sensitivity is not continuously affected by these issues and the appearance and effect-magnitude changes from time to time. (Koepken (2004)) has shown that

the occurrence of this anomaly can be detected by analysing the average IR intensity (over an entire image) over time or by cross-referencing with another satellite. However, requiring a complete image average means there is only 1 observation every 30 minutes, while the anomaly can continuously change and, in the meantime, the temperature of the Earth also changes (e.g. as night progresses across the Earth from the satellite's viewpoint). Therefore, this method for detection is not very sensitive or reliable. Cross-referencing measurements from the affected satellite with data from another satellite is unfortunately not

always possible. It is also preferable to have a detection concept that is independent of other data sources. Therefore, the selected detection approach models the effect of the anomaly as an (intensity) offset per scanline per image, calculated by

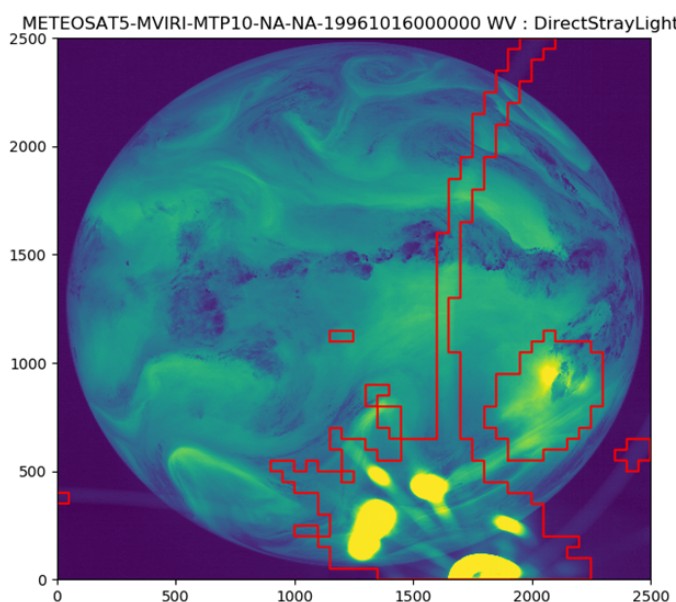

**Figure 9.** Segmentation result of the direct straylight anomaly detection.

least-squares optimization. The optimization makes two assumptions: i) the average intensity of a scanline in an image is equal to the average intensity of the same scanline in the preceding image; ii) the effects of the anomaly average out over N timeslots. The first assumption is in general valid for registered images without any anomaly. The second assumption is a direct consequence of a time-dependent stochastic process. Based on the results, we can conclude that the second assumption is valid

5 for the known optical hardware issues of Meteosat 5 and 6.

The bias for each scanline per image corresponding to this anomaly type can be calculated by solving a linear systems of equations, from which we define the following variables:

- X[i,j] = measured average intensity of scanline j from image i.

- B[i,j] = bias due to the instable optics anomaly of scanline j from image i

10 Assumption 1 results in the following equation:

$$X[i,j] - B[i,j] \approx X[i-1,j] - B[i-1,j] \Rightarrow X[i,j] - X[i-1,j] \approx B[i,j] - B[i-1,j]$$

Of course, these equations hold for every image $i$. Assumption 2 results in the following equation:

$$B[i,j] + B[i+1,j] + B[i+2,j] + ... + B[i+N,j] \approx 0$$





The equations for all images and scanlines can be stored in a matrix. As example a part of the matrix, where the anomaly is averaged out over 5 images with weight W:

$$
\begin{pmatrix}
-1 & 1 & & & & & & \\
& -1 & 1 & & & & & \\
& & -1 & 1 & & & & \\
& & & -1 & 1 & & & \\
& & & & -1 & 1 & & \\
& & & & & -1 & 1 & \\
& & & & & & -1 & 1 \\
W & W & W & W & W & & & \\
& W & W & W & W & W & & \\
& & W & W & W & W & W & 
\end{pmatrix}
\cdot
\begin{pmatrix}
B[i,j] \\
B[i+1,j] \\
B[i+2,j] \\
B[i+3,j] \\
B[i+4,j] \\
B[i+5,j] \\
B[i+6,j] \\
B[i+7,j]
\end{pmatrix}
\approx
\begin{pmatrix}
X[i+1,j] & - & X[i,j] \\
X[i+2,j] & - & X[i+1,j] \\
X[i+3,j] & - & X[i+2,j] \\
X[i+4,j] & - & X[i+3,j] \\
X[i+5,j] & - & X[i+4,j] \\
X[i+6,j] & - & X[i+5,j] \\
X[i+7,j] & - & X[i+6,j] \\
& 0 & \\
& 0 & \\
& 0 &
\end{pmatrix}
\tag{1}
$$

In our case, the linear system covers in total 21 consecutive timeslots, where the anomaly averages out over 5 timeslots and
5    the anomaly bias is modelled per 100 scanlines. The calculated bias per image per scanline can be stored as an array (like the left figure of Figure 10), to better see how bias per scanlines changes within an image or in time.

The calculated bias of images that have been affected by this anomaly will be, in general, close to one digital count. The detection of this anomaly in an image is based on the average magnitude of the calculated scanlines' biases in a particular image.

## 4    Results

For each anomaly type, a dedicated algorithm has been developed. The detection performance of the algorithm on the training-set (covering 2500 timeslots) has been manually verified. During this verification process, we noticed that algorithms found more anomalies in the training-set than were manually found. After inspection of the new detections, it appeared that in most cases the algorithms were correct. If we look at the overall detection performance of the algorithms on the basis of the training-
15   set dataset, 97.7% (2.3% missed cases) of the anomalies are successfully (true positive) detected. 2.7% of the detections are incorrect (false positive). The detection accuracy differs for the various anomaly types - several anomaly types are detected with 100% accuracy, while more difficult anomaly types are successfully detected in approximately 90% of the cases. We see that several anomaly types are related to certain satellites, and that images from Meteosat 2 and 3 are more often affected by anomalies than the newer satellites. We often see that one image is affected by multiple anomalies simultaneously.
20   After the verification of the training-set, all images from the MFG satellites have been processed and the anomaly detections stored in a separate database. An overview of the detected anomalies in the entire MFG dataset is presented in Table 4. The percentage of level 1.0 files that contain an anomaly of a certain type (affecting any of the channels) is shown separately per satellite. The MET6 dataset contains mostly RSS images, which explains some of the high anomaly rates. The database can be

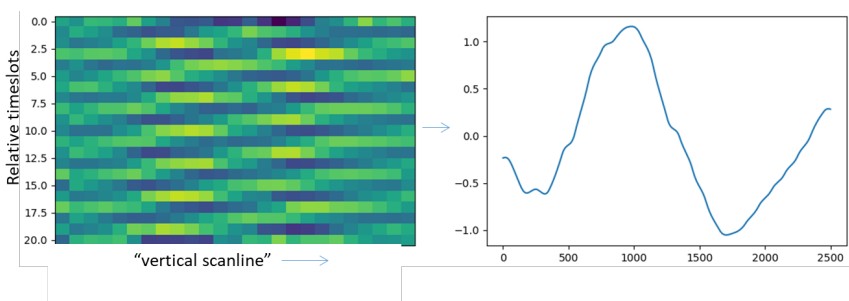

**Figure 10.** Example of the Meteosat 5 optical hardware issue in time, where the left image shows the anomaly's bias for consecutive images across the scanlines. The right image is for a particular timeslot and shows the anomaly's bias across the scanlines.

used to create statistical analyses about the anomaly distribution, or to filter images (or even pixels) for reprocessing campaigns or for specific and sensitive use cases such as cross-calibration.

## 5 Discussion and conclusions

Monitoring the quality of input data is of great importance in guaranteeing the correctness of the results of any analysis. For long-term studies, such as those on the Earth's climate, there is a need to combine data sets of various instruments, including those of early sensors that were not originally well quality-controlled. Later efforts to identify anomalies face multiple difficulties – loss of the original human expertise, limited documentation and datasets too large to assess manually in retrospect. A practical quality-assessment system must be based on automatic means and, rather than merely removing imperfect data from a period where observations are limited, must be able to support a wide variety of future uses by supplying detailed and precise information on the form and impact of anomalies. In addition, a uniform approach towards assessing the quality of data products is of great benefit to improving consistency over multiple sensors.





**Table 4.** Percentage of level 1.0 files containing anomalies in the images and metadata.

| CATEGORY | TYPE | MET2 | MET3 | MET4 | MET5 | MET6 | MET7 |
|---|---|---|---|---|---|---|---|
| artefact | east-west horizon misaligned | 0.1 | < 0.1 | 0.1 | < 0.1 | < 0.1 | < 0.1 |
| | overillumination | 32.9 | 34.9 | < 0.1 | < 0.1 | < 0.1 | < 0.1 |
| | tilted line | < 0.1 | < 0.1 | 3.8 | 3.9 | < 0.1 | < 0.1 |
| celestial body | celestial body: the Moon | 0.3 | 0.4 | 0.3 | 0.4 | 0.2 | 0.4 |
| | celestial body: undefined | 0.1 | 0.1 | 0.1 | < 0.1 | < 0.1 | 0.1 |
| corrupt or missing | completely black | 3.4 | 1.1 | 0.6 | < 0.1 | 2.5 | 0.4 |
| | corrupt file | < 0.1 | < 0.1 | < 0.1 | < 0.1 | < 0.1 | < 0.1 |
| | hanging scanline | 1.7 | 2.6 | 0.9 | 0.9 | 0.9 | 0.6 |
| | hanging scanline: no sub images | < 0.1 | < 0.1 | < 0.1 | < 0.1 | < 0.1 | < 0.1 |
| | incomplete image | 1.4 | 2.5 | 1.4 | 0.2 | 62.4 | 0.5 |
| | invalid signal | 100 | 100 | 0.1 | 0.1 | < 0.1 | 0.2 |
| | large black area | 3.8 | 4.0 | 2.7 | 2.8 | 2.3 | 1.1 |
| | large white area | 0.8 | 0.3 | < 0.1 | < 0.1 | < 0.1 | < 0.1 |
| | no sub images | 0.2 | 0.4 | 0.8 | 0.3 | 1.0 | 1.1 |
| hot pixel | hot pixel pattern 1 | < 0.1 | < 0.1 | 91.4 | 92.0 | 74.8 | 90.9 |
| | hot pixel pattern 2 | 0.4 | 0.2 | 4.7 | 13.7 | 2.9 | 5.0 |
| | hot pixel pattern independent | 40.6 | 5.0 | 18.1 | 24.2 | 2.3 | 9.4 |
| instable optics | instable optics | 0.4 | 0.3 | 0.2 | 2.7 | 10.2 | 0.2 |
| low SNR | low SNR: scanline | 16.2 | 1.3 | 0.3 | 0.1 | < 0.1 | 0.1 |
| raw data manipulated | background noise removed | 98.8 | 98.6 | 13.6 | < 0.1 | 0.1 | < 0.1 |
| | background noise removed / noise added | 91.1 | 10.9 | 0.7 | < 0.1 | 0.4 | < 0.1 |
| | the number of scanlines changed | 2.1 | 0.2 | < 0.1 | 0.1 | < 0.1 | < 0.1 |
| stray light | direct stray light | 3.9 | 5.6 | 4.8 | 6.8 | 1.9 | 4.7 |
| | indirect stray light | 6.1 | 4.7 | 4.9 | 6.4 | 2.2 | 6.1 |
| | reflection of the Moon | 0.1 | 0.2 | < 0.1 | < 0.1 | < 0.1 | < 0.1 |
| suspicious spectrum | suspicious spectrum | 6.9 | 44.9 | 60.4 | 2.2 | 0.1 | 1.8 |
| meta data | EFF position corrupt | 26.6 | 81.8 | 1.7 | 0.2 | < 0.1 | < 0.1 |
| | orbit position empty | 2.3 | 1.3 | 2.0 | 0.3 | 1.1 | 1.5 |
| | parameter empty | 100 | 100 | 100 | 17.3 | 63.9 | 1.8 |
| | start time: forward scan | < 0.1 | < 0.1 | < 0.1 | < 0.1 | < 0.1 | < 0.1 |
| | start time: southern horizon | < 0.1 | < 0.1 | < 0.1 | < 0.1 | < 0.1 | < 0.1 |
| | start time: start image | 70.0 | 0.2 | 0.1 | 0.4 | 62.7 | 0.3 |
| | start time: undefined | < 0.1 | < 0.1 | < 0.1 | 0.1 | < 0.1 | 0.1 |
| | value unexpected | 100 | 100 | 100 | 17.3 | 63.9 | 1.8 |



This paper describes a general method to screen an EO image database with a cumulative observation history of approximately 40 years. It has been shown that the method of using dedicated anomaly detection algorithms is sufficiently powerful to detect a wide array of anomalies, ranging from clear faults to subtle problems related to straylight that occur only under certain celestial constellations. The main challenge was to develop the methods such that the algorithms accurately detect the

images that are affected by the anomalies and, within the images, which areas are affected. With respect to the first objective, the Probability of Detection for affected images has been established at 97.7% and the False Alarm Rate of the method is 2.7%. The specificity within an affected image of the method is subjectively very good and most of the detection algorithms are able to highlight only those pixels that are affected.

The anomaly detection results for the full dataset of EUMETSAT's MFG satellites are stored in a dedicated database that

can be consulted to better understand the distribution of anomalies over the complete data set, and to filter the image data so that long-term analyses are being conducted on quality-controlled input data.

The anomaly detection system will be an essential part of the quality control system in future reprocessing and analysis work, and strengthens EUMETSAT's stewardship of the full MFG data archive by providing a consistent and data-based methodology for quality assessment. Although the anomaly detection algorithms have been tested on MFG data, it is believed

that the approach can be used for other similar geostationary satellite instruments as well, such as those on MSG (Schmetz and Menzel (2015)), GMS/MTSAT (Tabata et al. (2019)) and GOES (Considine (2006)), as no satellite-specific knowledge is needed to parameterize the detection algorithms.

*Author contributions.* Conceptualization, V. John, A. Bos, F. Liefhebber, J. Schulz; Formal analysis: F. Liefhebber, Investigation, F. Liefhebber; Methodology: F. Liefhebber, Project administration: S. Lammens, Software: F. Liefhebber, P. Brussee, F. Rüthrich, M. Grant; Validation:

J. Onderwaater, F. Liefhebber. Visualisation: F. Liefhebber; Supervision: J. Schulz, A. Bos; Writing – original draft: F. Liefhebber, V. John, A. Bos; Writing – review and editing: F. Liefhebber, A. Bos, V. John, F. Rüthrich, J. Onderwaater and M. Grant.

*Competing interests.* The authors declare no conflict of interest.



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
