# Peer review of "Automatic Quality Control of the Meteosat First Generation Measurements"

_Atmospheric Measurement Techniques, 2019_

## Referee Comment (RC1) · Jörg Trentmann (Referee) · 10 Sep 2019

The manuscript by Liefhebber et al. describes an automatic quality control procedure for the data provided by the first generation of Meteosat satellites, which have been in orbit between 1981 and 2017. Exploration of these data sets for climate analysis has not been the main purpose of the data collection at the time when this data was acquired, so no particular attention had been paid to the requirements related to climate data, e.g., the documentation of data anomalies.

The manuscripts describes methods to automatically detect so-called 'anomalies' in these geostationary satellite data; these anomalies should not be used in retrieval systems when deriving thematic climate data records from the satellite data. Multiple typical anomalies have been identified, each of them is individually detected. For three of the typical anomalies, details of the detection algorithm are presented in this manuscript, while no in depth information is provided for the detection of other anomalies.

The manuscript is clearly structured, well written and contains relevant information. I am recommending publishing this manuscript in AMT after my minor revisions are incorporated.

General: It would be most useful if the information on the anomalies detected in the satellite data would be accessible to users to avoid using anomalous satellite data when creating a climate data record. Please provide a list / a data base of the problematic satellite slots, e.g., as a supplement to this manuscript.

Please add the information on the satellite slots and channels (if appropriate) to all figures depicting satellite data, i.e., Figs. 2, 3,4, 5, 7, 8, 9, 10.

Page 1, line 17: Please change 'solar irradiation' to 'surface solar irradiance'.

Page 3, line 5 ff: Please add the information on the time it takes for the satellite to finish the forward scan. Also it might be of interest to the reader to have an idea on the overall size of the data set, so maybe the total size of the data set can be added in this section as well (it should be a pretty impressive data amount).

Page 4, line 14: Consider starting a new paragraph after '... limited dataset.', which begins with 'The remainder of this section...'

Section 3.1 / Figure 1: The creation of the training-set needs to be more clearly described. My understanding is that as a first step, '[...] samples were randomly selected, but with several constraints [...]' (p4, line 31); these samples (please specify the number of randomly selected samples) have been manually inspected '[...] to determine if it contains an anomaly.' (page 5, line 9). To improve the quality of the training data set (Is it intended that that training data set only contains problematic satellite slots?) an

'anomaly detection '-algorithm was applied to the training data set that either supports or falsifies the result from the manual inspection. The result of this 'automatic anomaly detection' is manually verified and, if an anomaly can be manually confirmed, this new case is "added to the training-set." (page 5, line 26). It is not clear to me how this description transfers into Figure 1. Does 'Ground truth' corresponds to the training-set? From the 'entire dataset' (all satellite slots, I assume) a subset is randomly selected (first box) and manually inspected (the 'subsetting' as suggested by the arrow should occur only after the manual inspection, if I am not mistaken). To the 'ground truth' slots (the 'training-set'?) the automatic anomaly detection as well as a manual verification is applied. In case the manual verification identifies an anomaly this slot is put back to the 'Ground truth' and tested again until the result has been verified.

Please add a box representing the 'Training-set' to Figure 1 (maybe it even is the outcome of the 'verification result'?); add information on the size of the different subsets, i.e, the 'subset' to start the generation of the training set as well we the size of the training set. Also please clearly specify whether the training set contains only anomalies or not.

Table 4: Please add the detection performance for each anomaly type to the table; also please add information on the affected satellite channel, if appropriate.

---

## Referee Comment (RC2) · Anonymous Referee #2 · 20 Sep 2019

AMT review

General comments

This is a clear, well-written paper describing anomaly-detection algorithms applied to Meteosat First Generation data that allow for quality control to screen out problematic values when using the data in climate applications. These algorithms could be usefully generalized to other geostationary sensors.

I recommend that this paper is published if the authors address my minor comments below.

Specific comments

[Figure]

Figure 10 - left-hand image should have a colour scale bar showing the magnitude of the bias.

Table 4 - MET6 has 62.4% "incomplete image" due to being configured for RSS as noted in the text. Why is the corresponding MET5 value only 0.2% when it was configured for RSS for ~5-10% of its operational life (according to Table 1)?

Table 4 - the "parameter empty" and "value unexpected" stats are identical possibly suggesting a strong overlap between these flags: is the former a subset of the latter case? Is it useful to maintain separate anomaly classifications for these?

More generally, when looking at Table 4, it would be very useful to provide some information about the relative importance of the different anomalies and their implications for the data. For example, what is the typical magnitude of the impact on the data, or what does "parameter empty" actually imply for the data (does it depend on which parameter was empty? Does an empty parameter invalidate an entire channel for a slot, or an entire slot?) At face value, MET2 and MET3 have 100% of slots flagged for 3 anomalies ("invalid signal", "parameter empty", "value unexpected"), and >98% of slots flagged for "background noise removed", but presumably this does not mean all the MET2 and MET3 data should be rejected? Of course, just because a slot is flagged, that does not indicate all data for all channels within the slot are affected, but some information about the impacts of the various anomalies would make these statistics easier to interpret, and would be essential for someone making use of these anomaly flags for quality control. This information could be provided in a separate table or in the text if it will not fit into table 3 or 4.

Technical comments

Page 5, lines 19/20, typo: remove duplicate "also".

---

## Short Comment (SC1) · 15 Oct 2019

The automatic quality control of the Meteosat measurements described in this manuscript will certainly be helpful for future analysis of MFG data. In my opinion the anomaly detection results are not just relevant for filtering the complete set of image data but constitute an important data set in their own right. One could for example imagine that a comparison of the stray light effects identified in flight with the predictions from optical models before launch will aid the design of new instruments without stray light anomalies. Also celestial bodies in the field of view could be interesting. Therefore it might be as desirable to have the capability to produce collections of data affected by certain anomalies as producing data sets containing no anomalies.

---

## Author Comment (AC1) · 31 Oct 2019

Referee 1

[GENERAL COMMENT]: The manuscript by Liefhebber et al. describes an automatic quality control procedure for the data provided by the first generation of Meteosat satellites, which have been in orbit between 1981 and 2017. Exploration of these data sets for climate analysis has not been the main purpose of the data collection at the time when this data was acquired, so no particular attention had been paid to the requirements related to climate data, e.g., the documentation of data anomalies. The manuscripts describes methods to automatically detect so-called 'anomalies' in these geostationary satellite data; these anomalies should not be used in retrieval systems

when deriving thematic climate data records from the satellite data. Multiple typical anomalies have been identified, each of them is individually detected. For three of the typical anomalies, details of the detection algorithm are presented in this manuscript, while no in depth information is provided for the detection of other anomalies. The manuscript is clearly structured, well written and contains relevant information. I am recommending publishing this manuscript in AMT after my minor revisions are incorporated.

[ANSWER]: Thank you very much for the positive comments.

[SPECIFIC COMMENTS] [COMMENT #1] It would be most useful if the information on the anomalies detected in the satellite data would be accessible to users to avoid using anomalous satellite data when creating a climate data record. Please provide a list / a data base of the problematic satellite slots, e.g., as a supplement to this manuscript.

[ANSWER]: EUMETSAT is currently undertaking an image reprocessing of the MVIRI images to produce new level 1.5 data in NetCDF format, which will be utilising the anomaly detection database. In the new level 1.5 files, there will be quality flags for users to identify anomalous images or data points and details on the anomalies.

Specific comments Please add the information on the satellite slots and channels (if appropriate) to all figures depicting satellite data, i.e., Figs. 2, 3,4, 5, 7, 8, 9, 10.

We will add the information where appropriate, according to the following table:

Figure filename channel

2 left METEOSAT3-MVIRI-MTP15-NA-NA-19900816133000 WV

2 right METEOSAT2-MVIRI-MTP10-NA-NA-19840215160000 IR

3, left METEOSAT4-MVIRI-MTP10-NA-NA-19911216110000 WV

3, center METEOSAT2-MVIRI-MTP10-NA-NA-19831016120000 WV

3, right METEOSAT2-MVIRI-MTP10-NA-NA-19810817223000 WV

4 left METEOSAT4-MVIRI-MTP10-NA-NA-19900415003000 WV

4 right METEOSAT5-MVIRI-MTP10-NA-NA-20060317213000 WV

5 left METEOSAT2-MVIRI-MTP10-NA-NA-19810817223000 WV

5 right METEOSAT4-MVIRI-MTP10-NA-NA-19901015013000 WV

7 left METEOSAT7-MVIRI-MTP15-NA-NA-20130717210000 WV

7 center METEOSAT7-MVIRI-MTP15-NA-NA-19991016003000 VIS

7 right METEOSAT4-MVIRI-MTP10-NA-NA-19900415003000 WV

8 left METEOSAT5-MVIRI-MTP10-NA-NA19961015300000 WV

8 center METEOSAT5-MVIRI-MTP10-NA-NA19961016000000 WV

8 right METEOSAT5-MVIRI-MTP10-NA-NA19961016300000 WV

9 METEOSAT5-MVIRI-MTP10-NA-NA19961016000000 WV

10 around METEOSAT5-MVIRI-MTP10-NA-NA-19960515120000 IR

[COMMENT #2] Page 1, line 17: Please change 'solar irradiation' to 'surface solar irradiance'.

[ANSWER]: Will be changed to 'surface solar irradiance'.

[COMMENT #3] Page 3, line 5 ff: Please add the information on the time it takes for the satellite to finish the forward scan.

[ANSWR] It takes 25 minutes to complete the forward scan. This information will be added to the document.

COMMENT #4] Also it might be of interest to the reader to have an idea on the overall size of the data set, so maybe the total size of the data set can be added in this section

as well (it should be a pretty impressive data amount).

[ANSWER]: The size of all 1.0 files that were being processed iss 47TB. This information will be provided in the document.

[COMMENT #5] Page 4, line 14: Consider starting a new paragraph after '. . . limited dataset.', which begins with 'The remainder of this section. . .'

[ANSWER]: The new paragraph will be inserted.

[COMMENT #6] Section 3.1 / Figure 1: The creation of the training-set needs to be more clearly described. My understanding is that as a first step, '[. . .] samples were randomly selected, but with several constraints [. . .]' (p4, line 31); these samples (please specify the number of randomly selected samples) have been manually inspected '[. . .] to determine if it contains an anomaly.' (page 5, line 9). To improve the quality of the training data set (Is it intended that that training data set only contains problematic satellite slots?) an 'anomaly detection '-algorithm was applied to the training data set that either supports or falsifies the result from the manual inspection. The result of this 'automatic anomaly detection' is manually verified and, if an anomaly can be manually confirmed, this new case is "added to the training-set." (page 5, line 26). It is not clear to me how this description transfers into Figure 1. Does 'Ground truth' corresponds to the training-set? From the 'entire dataset' (all satellite slots, I assume) a subset is randomly selected (first box) and manually inspected (the 'subsetting' as suggested by the arrow should occur only after the manual inspection, if I am not mistaken). To the 'ground truth' slots (the 'training-set'?) the automatic anomaly detection as well as a manual verification is applied. In case the manual verification identifies an anomaly this slot is put back to the 'Ground truth' and tested again until the result has been verified.

[ANSWER]: Apologies for the confusion. Indeed, Figure 1 is not correct. The 'Ground truth' as described in the figure should indeed read as 'training-set. We will change FIgure 1 by replacing the description in the 'Ground Truth' box into 'Training set'.

[COMMENT #7] Please add a box representing the 'Training-set' to Figure 1 (maybe it even is the outcome of the 'verification result'?); add information on the size of the different subsets, i.e, the 'subset' to start the generation of the training set as well we the size of the training set. Also please clearly specify whether the training set contains only anomalies or not.

[ANSWER]: With respect to the 'Training set' in Figure 1, please see our answer directly above. We will add the size of the training set to the description. And also specify whether only anomalies are contained in the 'Training set'.

———————————————————————

[Figure]

[Figure]

**Fig. 1.** Update Figure 1 (Training set issue)

[Figure]

---

## Author Comment (AC2) · 31 Oct 2019

Apologies, I have forgotten to address a comment:

[COMMENT]: Table 4: Please add the detection performance for each anomaly type to the table; also please add information on the affected satellite channel, if appropriate.

[ANSWER]: The overall Probability of Detection of the anomalies has been determined based on the results from the training set. The POD of each anomaly type has not been determined, because the number of occurrences in the training-set is considered to be too low to reliably calculate the POD. For some anomaly types the sensitivity level of the algorithm (should an image where an anomaly is vaguely visible be flagged or not) could be subject to end-user's preference, and as such the sensitivity level will affect the

[Figure]

POD and the false alarm rate of the algorithm. The simple anomaly categories, such as "corrupt or missing" and "hot pixel", the anomalies are detected in all cases. For the more complex anomaly types, such as "direct stray light" and "suspicious spectrum", the POD is around 90%.
* * *

---

## Author Comment (AC3) · 31 Oct 2019

Referee 2 This is a clear, well-written paper describing anomaly-detection algorithms applied to Meteosat First Generation data that allow for quality control to screen out problematic values when using the data in climate applications. These algorithms could be usefully generalized to other geostationary sensors. I recommend that this paper is published if the authors address my minor comments below.

[ANSWER]: Thank you very much for the positive comments.

Specific comments [COMMENT] Figure 10 - left-hand image should have a colour scale bar showing the magnitude of the bias.

[ANSWER] We agree that the left-hand image of Figure 10 with colour scale will provide

a more information to the reader, and therefore the figure is updated.

[COMMENT] Table 4 - MET6 has 62.4% "incomplete image" due to being configured for RSS as noted in the text. Why is the corresponding MET5 value only 0.2% when it was configured for RSS for âĹij5-10% of its operational life (according to Table 1)?

[ANSWER]

As shown in EUMETSAT Satellites History document (EUM/OPS/DOC/08/4698, link below) Met5 was doing RSS from 21/04/1997 to 03/07/1997. However, there are only a very few RSS data files available during that time.

https://www.eumetsat.int/website/wcm/idc/idcplg?IdcService=GET_FILE&dDocName=PDF_METEOSAT_PRIME_SATELI

[COMMENT] Table 4 - the "parameter empty" and "value unexpected" stats are identical possibly suggesting a strong overlap between these flags: is the former a subset of the latter case? Is it useful to maintain separate anomaly classifications for these? More generally, when looking at Table 4, it would be very useful to provide some information about the relative importance of the different anomalies and their implications for the data. For example, what is the typical magnitude of the impact on the data, or what does "parameter empty" actually imply for the data (does it depend on which parameter was empty? Does an empty parameter invalidate an entire channel for a slot, or an entire slot?) At face value, MET2 and MET3 have 100% of slots flagged for 3 anomalies ("invalid signal", "parameter empty", "value unexpected"), and >98% of slots flagged for "background noise removed", but presumably this does not mean all the MET2 and MET3 data should be rejected? Of course, just because a slot is flagged, that does not indicate all data for all channels within the slot are affected, but some information about the impacts of the various anomalies would make these statistics easier to interpret, and would be essential for someone making use of these anomaly flags for quality control. This information could be provided in a separate table or in the text if it will not fit into table 3 or 4.

We agree with the reviewer that some of the anomalies that are flagged to a very high percent will not make the data unusable. For example, the "background noise removed" anomaly for MET2 and MET3 will only hinder the recalculation of instrument noise (computed as the space corner noise) or space count values for these instruments. But operationally computed values for these parameters are already available in the data and could be used. The images themselves are not affected by the removed background noise. This will only affect when these images have to be recalibrated as described in Ruethrich et al, 2019.

EUMETSAT is currently undertaking an image reprocessing of the MVIRI images to produce new level 1.5 data in NetCDF format, which will be utilising the anomaly detection database. In the new level 1.5 files, there will be quality flags for users to identify anomalous images and information on which data and metadata is affected.

Rüthrich, F., V. O. John, R. A. Roebeling, R. Quast, Y. Govaerts, E. Wooliams, and J. Schulz (2019) Climate Data Records from Meteosat First Generation Part III: Recalibration and Uncertainty Tracing of the Visible channel on METEOSAT 2-7 using Reconstructed, Spectrally Changing Response Functions, Remote Sens., 11, 1165, https://www.mdpi.com/2072-4292/11/10/1165.

[COMMENT] Technical comments Page 5, lines 19/20, typo: remove duplicate "also"

[ANSWER]

One of the 'also' words will be removed.

---

## Author Comment (AC4) · 31 Oct 2019

The automatic quality control of the Meteosat measurements described in this manuscript will certainly be helpful for future analysis of MFG data. In my opinion the anomaly detection results are not just relevant for filtering the complete set of image data but constitute an important data set in their own right. One could for example imagine that a comparison of the stray light effects identified in flight with the predictions from optical models before launch will aid the design of new instruments without stray light anomalies. Also celestial bodies in the field of view could be interesting. Therefore it might be as desirable to have the capability to produce collections of data affected by certain anomalies as producing data sets containing no anomalies.

[Figure]

[ANSWER] Thank you very much for your thoughtful comments. We agree that the use cases for the tool extend indeed beyond excluding anomalous data for re-processing of long-term data.
* * *

---

## Author Response (AR2)

**Document**

**amt-2019-249** Submitted on 19 Jun 2019

**Automatic Quality Control of the Meteosat First Generation Measurements**

Freek Liefhebber, Sarah Lammens, Paul Brussee, André Bos, Viju O. John, Frank Rüthrich, Jacobus Onderwaater, Michael G. Grant, and Jörg Schulz

**Editor's comment:**

Associate Editor Decision: Publish subject to minor revisions (review by editor) (03 Dec 2019) by Andrew Sayer
Comments to the Author:
Dear authors,

Thank you for submitting your revised manuscript; after reading it and your Response to Reviewers through I feel that you have addressed the reviewers' technical questions well through these revisions and the addition of the Supplement.

However, as one minor point, the figures are all missing units (and aside from Figure 10, colour scales). As the main interpretation of Figures 2, 3, 4, 5, 7, 8, and 9 is qualitative and not quantitative, I don't think a colour scale is necessary for these Figures. However it would be useful to the reader to add some interpretation in the captions: I infer this is digital counts being mapped linearly (with purple through yellow being low to high), but this should be stated directly somewhere (e.g. in the Figure 2 caption, and then add a statement in later captions referring back to Figure 2). Similarly, both panels of Figure 10 need the units stated (and the right-hand panel needs axis labels, even though the x-axis is described in the caption).

Other than that, I am happy to accept your manuscript for publication in AMT.

Best wishes,

Andrew

**Reply**

[revised manuscript text omitted]